# Peer review of "The Clinical Guiding Role of the Distribution of Corneal Nerves in the Selection of Incision for Penetrating Corneal Surgery in Canines"

_vetsci, 2021, doi:10.3390/vetsci8120313_

Round 1
Reviewer 1 Report
This paper measured the number of corneal nerve roots invading from the corneal liimbus of dogs by direction. For intraocular surgery, an incision needs to be made in the cornea, which results in corneal nerve amputation, inducing various eye diseases. The results obtained from this study suggest the location and orientation of the incision, which may reduce the number of corneal nerve roots to be amputated during intraocular surgery.
Abstract: Please refer to the following comments.
L10-16: The background is too long, but the material is not described at all. At least the following information about the material needs to be included: ”39 corneas from 23 canines, including 12 male canines and 11 female canines.”
L22 describes “The nerve roots in both eyeballs were symmetrically distributed.”, whereas L18-19 read “There was no significant difference in the number of nerve roots at the corneoscleral limbus between different zones (F = 1.983, p = 0.082), and the nerve roots at the corneoscleral limbus (mean value, 24.43; 95% CI, 23.43-25.42) were evenly distributed. ” Since these descriptions are redundant, it would be advised to remove either of them.
Materials and Methods: Please refer to the following comments.
L108: Does “corneal endodermis and posterior elastic layer” mean “Descemet's membrane” alias “posterior limiting layer”? It would be advised to use the term according to standard ophthalmic terminology.
L110: Regarding “6 quadrants (Figure 1)”, it is necessary to reconsider how to label the zones. In order to compare the left and right corneas, if you label the zones clockwise from a to f for the right eye, you must put the labels counterclockwise for the left eye.
L116: “Fresh lemon juice” is an inappropriate term in scientific papers; one cannot perform a reproduction experiment from this information. It would be advised to use a more scientifically precise expression.
L123: Regarding “fixing solution”, the composition is unknown. Please specify the contents.
Results: Please refer to the following comments.
L141-143: “In this experiment, 39 corneas from 23 canines were successfully stained, including 141 12 male canines and 11 female canines (21 corneas from the left eyeballs and 18 corneas 142 from the right eyeballs).” It is advised to move this part to Materials and Method section. In addition, please clarify the age of each canine. Please also refer to the comment in the Discussion where the necessity of age specification is also commented.
Discussion: Please refer to the following comments.
It will be of interest to the reader to compare this paper’s results with those already obtained for humans. Reference 26 (He, J., et al., 2010. "Mapping the entire human corneal nerve architecture." Exp Eye Res 91: 513-523.) observed human corneal nerves and provided detailed discussion. The paper by He et al. pointed out the followings.
- Epithelial nerve density and terminal numbers were higher in the center of the cornea, rather than the periphery.
- There were no differences in epithelial nerve density between genders, but there was a progressive nerve density reduction concomitant with aging
It is meaningful in comparative ophthalmology to make comparisons between the findings obtained for humans and the findings obtained for dogs in this study. This is the reason why I asked for information on the target ages earlier. However, the following points must be taken into consideration when making comparisons: this paper divided a cornea into 6 zones, whereas He et al. divided one into 4 quadrants; the subjects of this paper were 8 months to 3 years old, while those of He et al. were mainly 70 years old and over.
The authors described in detail the difficulties with gold chloride staining and how they overcame the difficulties. Unfortunately, no results were given. If it is not the main point of this paper, it would be desirable to delete this part. Or the authors should provide the results if they wish to retain this part in the manuscript.
Tables and Figures: The contents of some figures may be too scarce. Please reduce the number of figures by integrating some of them.
Figure 1 and 3: These two figures may be made into one. By presenting the left and right sides of the same animal side by side, you may create an attractive figure. Insert the name of each zone from a to f. Please refer to the comment for Materials and Methods on how to name each zone on the left and right eyes.
Figure 4 and 5: The contents of the both figures are well explained in the text or in the table. Neither of the figure may be essential. In general, ophthalmology papers show OD on the left and OS on the right. When keeping these figures, it would be better to switch the left and right figures.
Figure 4: The horizontal axis indicates “Total number of corneal nerve roots”, but the numbers are too large. There seems to be an error in the unit or numerical value. Please confirm if it is the sum of the measurement results of 39 corneas.
Author Response
Please see the attachment.
Thank you for your letter and for the reviewers’ comments concerning our manuscript entitled “The Clinical Guiding Role of the Distribution of Corneal Nerves in the Selection of Incision for Penetrating Corneal Surgery in Canines” (ID: vetsci-1454802).

Reviewer 2 Report
The study is of interest and is in line with the scope of the journal.
The paper is generally well structure.
the abstract cleary indicates the work objective, methodology and result of the study .
the methodology is well articulated and the description of the case is adequately presented.
in my opinion the manuscript could be accepted for publication.
Author Response

(The authors gave the same response as above.)

Reviewer 3 Report
The authors provide a very interesting paper, shedding the light into a yet-poor-considered filed, those of nerve damage in case of corneal injury or surgery.
The paper is well written and the study design is robust. The reference list is updated and in the paper are provided almost the most useful iniformation on this topic.
I found only minor areas to be elucidated, as detailed below:
Introduction section: very informative but too long and difficult to follow. I advise to shorten
Page 2, line 60. Which study is those refers by the Authors?
2.1. Animal and Tissue Sample Collection:
what were the breeds of dogs donors of cornea?
Regarding clinical examination, when it was performed? When they were still alive or just died? The timing of ophthalmic examination is crucial in this case, since this would change the normality pattern showed by the cornea.
Author Response

(The authors gave the same response as above.)

Round 2
Reviewer 1 Report
The authors have satisfactorily responded to all my questions and made the necessary changes to the manuscript.